# Hypofibrinolysis induced by tranexamic acid does not influence inflammation and mortality in a polymicrobial sepsis model

**Yzabella Alves Campos Nogueira[1], Loredana Nilkenes Gomes da Costa[1,2], Carlos Emilio Levy[1], Fernanda Andrade Orsi[1], Franciele de Lima[1], Joyce M. Annichinno-Bizzacchi[1,3], Erich Vinicius De Paula[1,3]***

**1** School of Medical Sciences, University of Campinas, Campinas, SP, Brazil, **2** Federal University of Piaui, Parnaiba, PI, Brazil, **3** Hematology and Hemotherapy Center, University of Campinas, Campinas, SP, Brazil

* erich@unicamp.br

**Data Availability Statement:** All relevant data are within the manuscript and its Supporting Information files.

## Abstract

The biological relevance of fibrinolysis to the host response to sepsis is illustrated by pathogens such as *S. pyogenes* and *Y. pestis*, whose virulence factors are proteins that challenge the balance between pro- and anti-fibrinolytic factors of the host, and by the consistent finding of hypofibrinolysis in the early stages of sepsis. Whether this hypofibrinolytic response is beneficial or detrimental to the host, by containing the spread of pathogens while at the same time limiting the access of immune cell to infectious foci, is still a matter of debate. Tranexamic acid (TnxAc) is an antifibrinolytic agent that is being increasingly used to prevent and control bleeding in conditions such as elective orthopedic surgery, trauma, and post-partum-hemorrhage, which are frequently followed by infection and sepsis. Here we used a model of polymicrobial sepsis to evaluate whether hypofibrinolysis induced by TnxAc influenced survival, tissue injury and pathogen spread. Mice were treated with two doses of TnxAc bid for 48h, and then sepsis was induced by cecal ligation and puncture. Despite the induction of hypofibrinolysis by TnxAc, no difference could be observed in survival, tissue injury (measured by biochemical and histological parameters), cytokine levels or pathogen spread. Our results contribute with a new piece of data to the understanding of the complex interplay between fibrinolysis and innate immunity. While our results do not support the use of TnxAc in sepsis, they also address the thrombotic safety of TnxAc, a low cost and widely used agent to prevent bleeding.

## Introduction

Coagulation activation is currently regarded as part of the host immune response to pathogens [1,2]. Evidences from animal models of sepsis suggest that pathways that lead to thrombin and fibrin formation contribute to pathogen clearance by both containing the spread of pathogen foci, and by generating antimicrobial peptides from cleaved coagulation proteins [3]. In addition, several lines of evidence suggest that hypercoagulability is also responsible for part of the

**Funding:** This work was funded by Coordenação de Aperfeiçoamento de Pessoal de Nível Superior - Brasil (CAPES) (www.capes.gov.br) - Finance Code 001 to YACN and was also funded by Sao Paulo Research Foundation (www.fapesp.br) grants # 2016/14172-6 and 2014/0984-3; and CNPq Brazil (www.cnpq.br) grant # 309317/2016; and FAEPEX-UNICAMP to EVDP. The funders had no role in study design, data collection and analysis, decision to publish, or preparation of the manuscript.

**Competing interests:** The authors have declared that no competing interests exist.

tissue damage associated with sepsis in cases associated with disseminated intravascular coagulation (DIC) [4].

While a lot of work has been devoted to understand the role of pro- and anticoagulant factors in the pathogenesis of sepsis, much less is known about how fibrinolysis participate in this complex process [5,6]. The relevance of fibrinolysis during sepsis is suggested by the consistent finding of hypofibrinolysis induced by PAI-1 release in the early stages of sepsis [1,7,8], and by examples of pathogens that use the cooption of fibrinolytic pathways as virulence factors [9,10].

Tranexemic acid (TnxAc) inhibits fibrinolysis by competitively inhibiting the binding of plasminogen to fibrin, thereby inducing a state of hypofibrinolysis that shifts the hemostatic balance towards a procoagulant state [11]. TnxAc has been classically used in the treatment of bleeding in patients with congenital or acquired bleeding disorders. More recently, use of this agent has increased due to the confirmation of its efficacy in settings such as elective orthopedic surgery [12], trauma [13] and post-partum-hemorrhage [14]. Therefore, exploring the effects of TnxAc on sepsis could contribute not only to improve our understanding about the interplay of fibrinolysis and inflammation, but also to gain insights on its thrombotic safety in patients at-risk for sepsis. Here we investigated the effect of TnxAc on a polymicrobial sepsis model, focusing on sepsis severity, organ damage, inflammatory response and pathogen clearance.

## Methods

### Animal model of polymicrobial sepsis

All procedures were approved by the Ethical Committee for Animal Experiments (CEUA) of University of Campinas under protocol 3432–1. C57BL/6J male mice, 8-weeks old, were obtained from CEMIB (University of Campinas, Campinas, SP, Brazil) and used in the experiments. The cecal ligation and puncture (CLP) sepsis model was used as previously described [15]. Briefly, mice were anesthetized with subcutaneous ketamine 80–100 mg.Kg$^{-1}$ and xylazine 5–15 mg.kg$^{-1}$. Through a left lateral laparotomy with about 50 mm, the cecum was identified and externalized. Silk 4.0 suture was used to occlude approximately 50% of the apical portion, then the occlude portion were transfixed for a 30G needle (Labor Import$^{®}$). After cecum reallocation. the abdominal wall and skin were synthesized. All mice received 1mL of subcutaneous saline as volume replacement, and subcutaneous injection of tramadol hydrochloride 0.05 mg.kg$^{-1}$ every 12 hours for analgesia until the time of euthanasia or death. Sample collection was performed in mice anesthetized in an isoflurane chamber. When indicated, euthanasia was performed with an overdose of isoflurane.

### Tranexamic acid (TnxAc) treatment

Mice were treated with TnxAc (Transamin, Laboratorios Pierre Fabre, Areal, RJ, Brazil) at two different doses (100mg/Kg or 600mg/Kg). The drug was diluted in sterile saline 0.9% and injected intraperitoneally every 12 hours, starting 2 days before sepsis induction, until the end of the experiment. Doses were arbitrarily chosen to span the range of doses used in previous studies using TnxAc in mice [16–18], with the 100mg/Kg dose more closely resembling regimens used in humans, and the 600mg/kg dose being at least one order of magnitude above clinically-used regimens. A sham group was treated with intraperitoneal injections of saline in the same regimen. Before sepsis experiments were initiated, the effect of TnxAc on fibrinolysis and on coagulation activation was assessed in mice treated with these regimens for 48 hours (described below).

## Survival after CLP induction

Survival was recorded by one investigator every 12 hours for 7 days, after CLP induction. In addition, clinical parameters were assessed daily by an investigator using a standardized score for murine sepsis (M-CASS), which considers the following markers: fur aspect, activity, posture, behavior, chest movements, chest sounds and eyelids, ranked form 1 to 4 [19]. In order to comply with the use of humane endpoints, we predefined criteria for euthanasia of animals presenting signs of severe distress: animals were daily evaluated by an investigator with a degree in veterinary medicine (Y.A.C.N) for clinical signs of sepsis using the M-CASS assessment score, and animals presenting labored breathing with gasps, coupled with absent righting reflex (turn to ventral decubitus) were immediately euthanized. The assessment for severe distress was also performed at each 12-hour survival assessment, totalizing at least 3 assessments per 24-hour period. In addition, all animals received 12-hour doses of analgesia, as previously stated. At the end of the observation period all surviving mice were euthanized. During these assessments, the experimental group of each mice was blinded.

## Sample collection and processing

Samples of whole blood, peritoneal fluid, and organs (liver, kidney and lung) were collected to assess hematological, biochemical, inflammatory, microbiological and histological parameters as described below, 24 hours after CLP induction. Whole blood was collected from the inferior vena cava into sterile syringes, and transferred into tubes containing anticoagulants (sodium citrate 3.8%; 1:9 proportion) or no anticoagulants. Platelet poor plasma was obtained by centrifugation of anticoagulated whole blood at 1,800g at 22˚C, for 15 minutes; serum was separated from blood left to clot at room temperature for 30 minutes by centrifugation at 1,000g for 10 minutes. Aliquots were then stored at 80˚C until analysis. For the collection of peritoneal fluid, the peritoneal cavity was exposed through a laparotomy incision and the cavity was washed with 0.5 mL of saline solution and the wash was collected using a sterile syringe. Finally, liver, kidneys and lungs were harvested for immediate microbiological analyses, or processed for histological analysis as described below.

## Hematological and biochemical analyses

Hemoglobin (Hb), total leukocyte and platelet counts were performed in an automatic hematology analyzer (Cell Dyn 1700 System, Abbot Diagnostics, Santa Clara, USA). Aspartate aminotransferase (AST), alanine aminotransferase levels (ALT), creatinine, blood urea nitrogen (BUN), creatinekinase (CK) and lactate dehydrogenase (LDH) were measured in an automated biochemistry analyser (Architect, Abbott Diagnostics, Abbott Park, IL, USA) in serum samples.

## Hemostasis assays

Thrombin-antithrombin (TAT) levels were measured in platelet poor plasma using a commercial immunoassay method (TAT Complexes Mouse Elisa Abcam® Kit). The euglobulin lysis time (ELT) was performed according to previous descriptions, adapted to our laboratory [20,21]. Briefly, 100μL of plasma was diluted with 1.8mL distilled water at 4˚C. To induce euglobulin fraction precipitation, acetic acid 0.25% (Merck®) was added to pH acidification (pH = 5.9) and the solution was incubated for 30 minutes at 4˚C. Tubes were sealed with parafilm M® and transferred to a centrifuge (Eppendorf® Centrifuge 5810R model), 1,800g at 4˚C for 10 minutes. The supernatant was discarded by inversion and the inner surface of the tubes dried with filter paper. Tubes were conditioned in a beaker of ice and water and the precipitate

was resuspended in 200 μl of Tris Tween 80 buffer at 0.1%. After resuspension, tubes were transferred to water bath at 37˚C. In 30 seconds, 100μl of bovine thrombin at 10U (Hemosil®) and calcium chloride (Merck® 0.0025M) were added. Clot dissolution was evaluated by visual inspection every 15 minutes. The time until complete clot dissolution was recorded as lysis time.

### Inflammatory makers

Levels of inflammatory cytokines were measured using a customized Magnetic Luminex® Screening Assay specific for mouse proteins (Mouse Premixed Multi-Analyte Kit) (R&D, Minneapolis, MN, USA) in plasma samples.

### Microbiological analyses

The number of colony forming units (CFU) was measured in samples of peritoneal fluid, tissue homogenates or whole blood samples plated in serial dilutions to optimize manual counting. Samples used in the experiments consisted of 10μl of whole blood or peritoneal fluid diluted in 1ml sterile saline; and liver or kidney homogenates diluted (1:100 v/v) in sterile saline. 5μl of each sample were seeded on 5% sheep blood agar plates (PlastLabor®, Brazil) and incubated at 37˚C for 24h, after which bacterial growth was assessed by manual counting of colonies, by an observer that was blinded to the experimental group of each sample. Results are expressed as CFU per mL.

### Histological analyses

Lungs and kidneys (both right) were stored in histological cassettes and fixed in buffered formalin 10% (Sigma®). After 24 hours the cassettes were included in paraplast resin (Sigma®), and serial sections were stained with hematoxylin and eosin (HE). Slides were coded to preclude groups identification analyzed by two blind observers for the presence of microvascular thrombosis. Microvascular thrombosis was defined as positive if at least two microvascular thrombi were observed at the same field.

### Statistical analysis

Data are expressed as means ± SEM or medians and range, as detailed in each section. Continuous variables were compared using the Mann-Whitney, Kruskal-Wallis or Anova tests, according to variable distribution and number of groups. All analyses were corrected for multiple comparisons. Survival curves were compared by the log-rank test. Differences were considered statistically significant if $P \leq 0.05$. The number of mice used in each experiment was calculated to obtain a power of 80% with a type I error of 0.05 to detect differences of at least two standard deviations, and ranged from 13 to 29 mice per group in survival curves, 6 to 11 in hemostasis assays, 5 to 12 in biochemical assays, 6 to 16 for inflammatory markers, and 12 to 15 in bacterial burden experiments. All statistical analyses were performed with the Graph-Pad Prism Software v 7.0 (GraphPad Prism Software Inc. San Diego, California, USA).

## Results

### TnxAc induces a hypofibrinolytic state in mice

Prior to sepsis experiments, mice were treated with TnxAc for 48 hours (under the same conditions used in the sepsis experiments) to confirm whether TnxAc was capable of inducing hypofibrinolysis. As shown in Fig 1A, both doses of TnxAc induced mild but significant increases in TAT levels. More importantly, we confirmed that TnxAc was capable of inducing

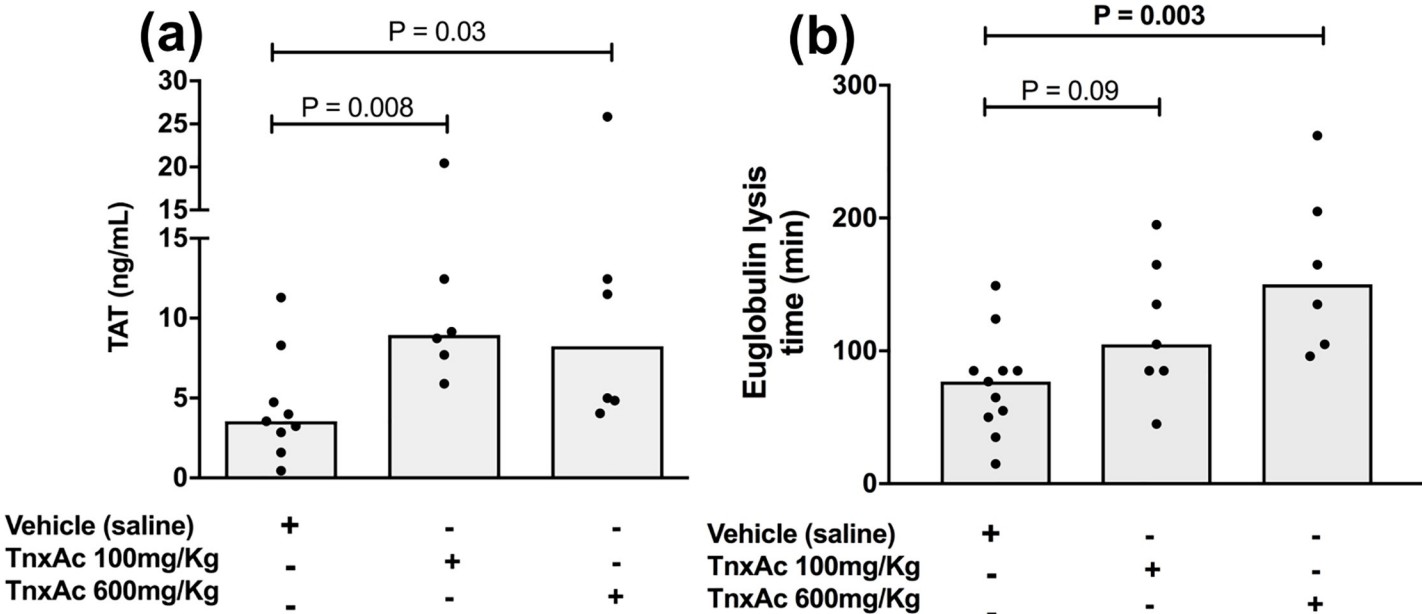

**Fig 1. Effect of TnxAc on coagulation and fibrinolysis parameters.** Mice were treated with the indicated doses of TnxAc ip, bid, for 48 hours. Platelet poor plasma was obtained from the inferior vena cava for analyses. (a) Plasma levels of thrombin-antithrombin (TAT) complexes and (b) the euglobulin lysis time are shown for TnxAc and vehicle treated mice are shown; n = 6–11 per group; Kruskal-Wallis test.

a hypofibrinolytic state in mice, with a dose-response effect (Fig 1B). In addition, no difference could be observed in Hb, leukocyte or platelet counts (Table 1).

## Hypofibrinolysis induced by TnxAc does not affect sepsis severity

We next evaluated whether hypofibrinolysis induced by TnxAc would influence mortality and severity in murine experimental sepsis. Mice treated with TnxAc at both doses (100mg/kg or 600mg/kg) did not present any signficant change in mortality (Fig 2). In order to identify more subtles differences in sepsis severity, mice were also evaluated using a clinical severity score, which also did not show any significant difference (Fig 3). Of note, one mouse was euthanized in the 600mg/kg TnxAc group experimental group due to the presence of signs of severe distress, according to predefined criteria, assessed by a blinded investigator.

## Hypofibrinolysis induced by TnxAc does not increase biochemical markers of tissue damage during polymicrobial sepsis

Levels of classical markers of liver injury (AST and ALT) and kidney dysfunction (creatinine and BUN) damage, as well as more general biomarkers of tissue damage (CK and LDH) did not differ between TnxAc and vehicle-treated mice irrespective of the dose used (Fig 4).

**Table 1. Effect of TnxAc on hematological parameters.**

| Parameters | Vehicle | 100mg/Kg | 600mg/Kg | *P |
|---|---|---|---|---|
| **Hb (g/dL)** | 13.1± 1.8 | 12,5 ± 2,07 | 10,1 ± 1,8 | 0.38 |
| **Platelets (*10^9/L)** | 902.7 ± 289.1 | 1215 ± 273,2 | 914 ± 362,9 | 0.57 |
| **Leukocytes (*10^9/L)** | 3.47 ± 0.84 | 5.17 ± 3.30 | 3.64 ± 1.06 | 0.67 |

* Krukal-Wallis test

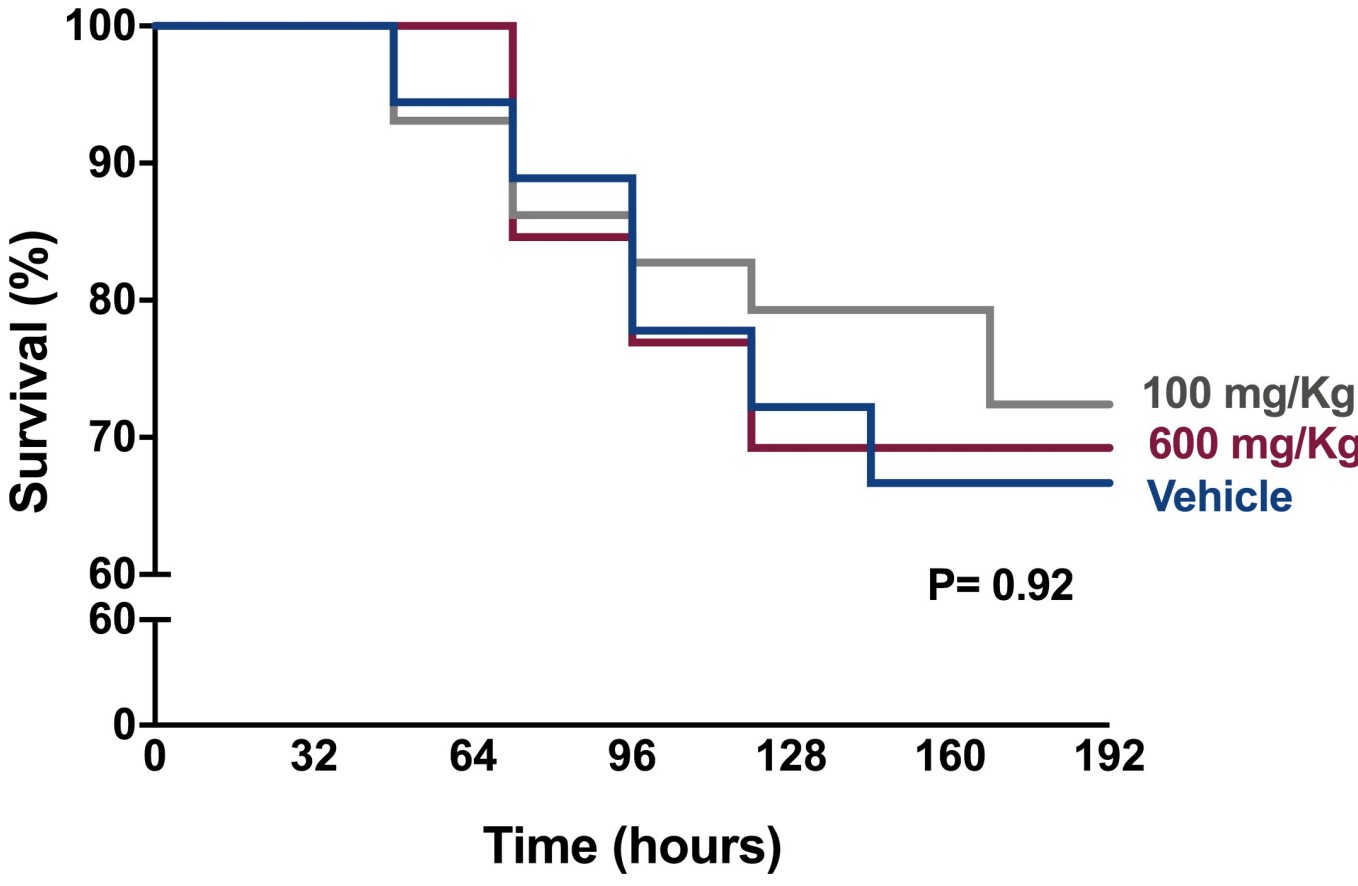

**Fig 2. Survival during polymicrobial sepsis.** Kaplan-Meier curves depicting survival of mice treated with TnxAc 100mg/kg bid (n = 13) or 600mg/kg bid (n = 18) initiated 48h before sepsis induction, for up to 7 days; Vehicle (n = 29); Log-rank test.

### Hypofibrinolysis induced by TnxAc is not associated with increased microvascular thrombosis

No evidence of increase microvascular thrombosis could be observed in kidneys or lungs of TnxAc-treated mice 24 hours before sepsis induction, when compared to the vehicle-treated group (Table 2).

### Hypofibrinolysis induced by TnxAc does not increase levels of inflammatory markers during polymicrobial sepsis

No statistically significant difference could be observed in levels of inflammatory mediators in plasma of mice treated with TnxAc at either dose, when compared to vehicle-treated mice (Fig 5).

### Hypofibrinolysis induced by TnxAc is not associated with significant decreases of bacterial dissemination

Bacterial clearance was studied in whole blood, peritoneal fluid, liver and kidneys. No statistically significant differences were observed in bacterial load in any of these tissues, at either tested doses (Fig 6). Yet, we could observe a trend towards lower bacterial burden in liver of

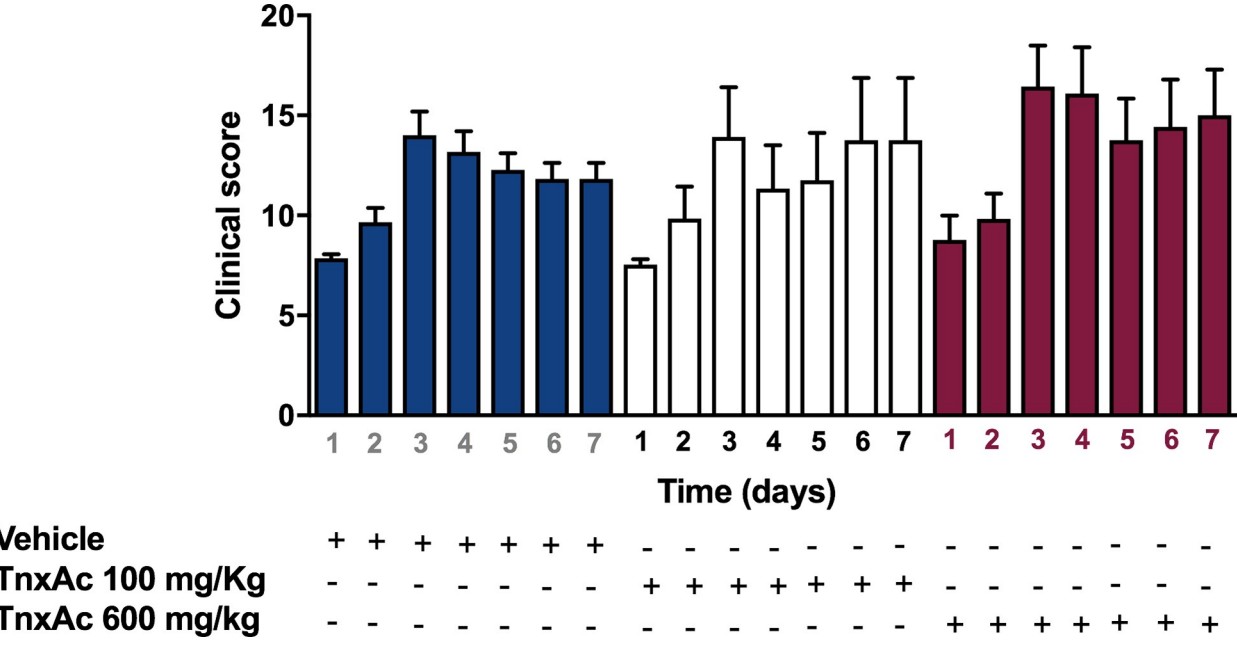

**Fig 3. Clinical score of the murine experimental sepsis.** Bars indicate daily mean clinical scores (and SEM) of mice treated with TnxAc at 100 mg/Kg, 600 mg/kg or vehicle (each bar represents one day of the follow-up, with data from mice that were remained alive at each time-point). No statistically significant difference could be observed between groups for each day separately (Kruskall-Wallis test with Dunn's multiple comparison test, comparing each time point between the three experimental groups).

mice treated with the highest dose of TnxAc (P = 0.052) (Fig 6C), which reached statistical significance when all TnxAc-treated mice were compared with vehicle-treated mice (S5 Fig).

## Discussion

The relevance of the fibrinolytic system to the pathogenesis of infectious diseases has been recognized for several decades [22], and is illustrated by pathogens that evolved virulence factors which are proteins capable to locally activate fibrinolysis, thereby facilitating pathogen evasion [9,10]. Here we used a pharmacological approach to inhibit fibrinolysis and a model of polymicrobial sepsis to gain further insights into the role of fibrinolysis in the pathogenesis of polymicrobial sepsis. Our main results were that hypofibrinolysis induced by TnxAc does not affect the clinical course of sepsis nor its subclinical markers of tissue damage and inflammation, and is not associated with statistically significant changes in pathogen spread.

Hemostasis and inflammation are seen as two highly interconnected processes, with sepsis being the clinical condition in which this association is more evident [23,24]. In the last decades, several laboratories have been trying to define whether coagulation activation is beneficial or detrimental during sepsis, with results pointing to both possibilities [3]. Fibrin deposition was the first compartment of hemostasis whose effect on sepsis was evaluated. In a rat peritonitis model, fibrin clots were associated with lower early mortality, but with increased abscess formation, leading to the hypothesis that fibrin formation would be protective in the early stages of sepsis (by trapping bacteria), but detrimental in later time points (by shielding pathogens from immune cells) [25]. This hypothesis was not confirmed in a randomized clinical trial in which radical peritoneal debridement failed to improve outcomes in patients with established bacterial peritonitis [26], nor in a more recent study that explored a similar concept using recombinant tissue plasminogen activator (t-PA) [27]. The importance of the fibrinolytic

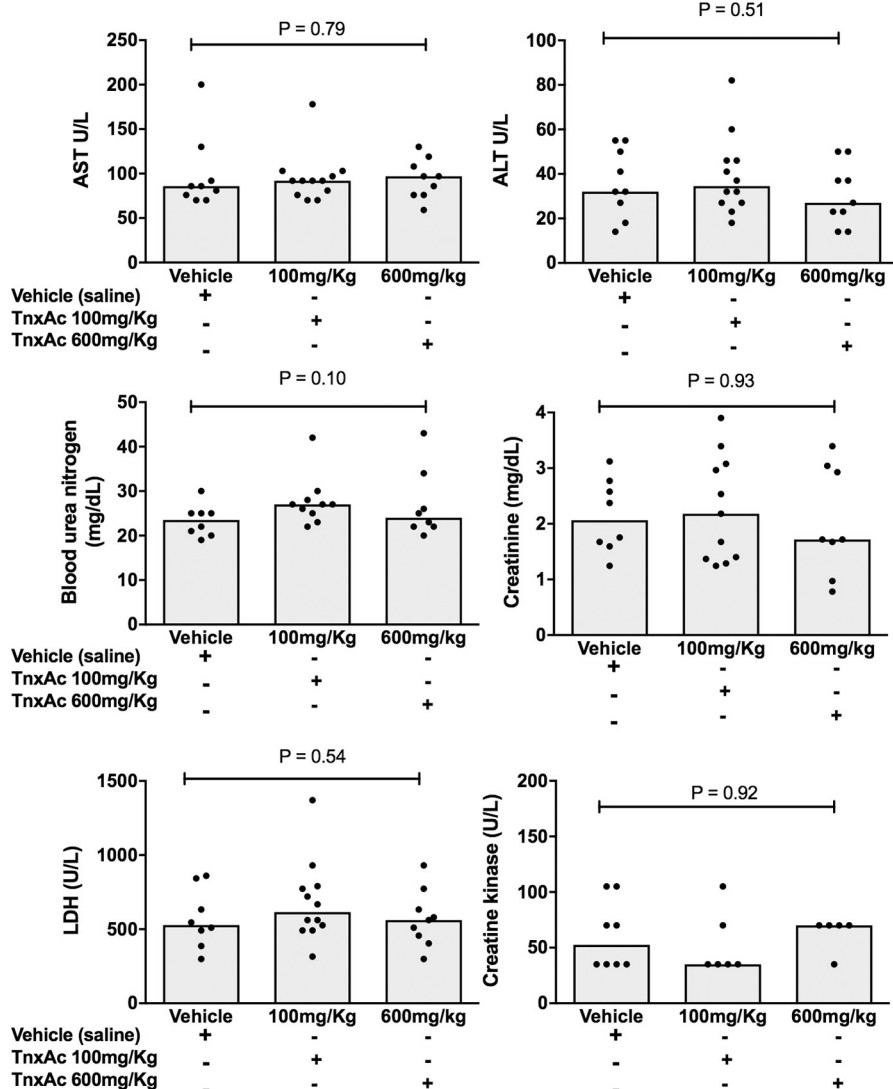

**Fig 4. Biochemical markers of tissue damage.** Median serum levels of (a) AST, (b) ALT, (c) BUN, (d) creatinine, (e) LDH and (f) CK in mice treated with TnxAc or vehicle were measured 24 hours after sepsis induction, and no differences were observed between groups; n = 5–12 per group; Kruskal-Wallis test.

system in the host response to sepsis is also supported by the early release of an anti-fibrino-lytic factor, plasminogen activator inhibitor 1 (PAI-1), in human endotoxemia [28], coupled by the consistent association of PAI-1 levels with sepsis mortality [7,8]. Accordingly, the role of other proteins involved in fibrinolysis in the pathogenesis of sepsis has been extensively

**Table 2. Frequency of microvascular thrombosis.**

| Microvascular thrombosis | Vehicle | 100mg/Kg | *P | 600mg/Kg | **P |
|---|---|---|---|---|---|
| **Kidneys, n** | 4/10 | 3/9 | NS | 4/10 | NS |
| **Lungs, n (%)** | 1/9 | 0/9 | NS | 0/8 | NS |

* Fisher's exact test vehicle x 100mg/Kg

** Fisher's exact test vehicle x 600mg/kg

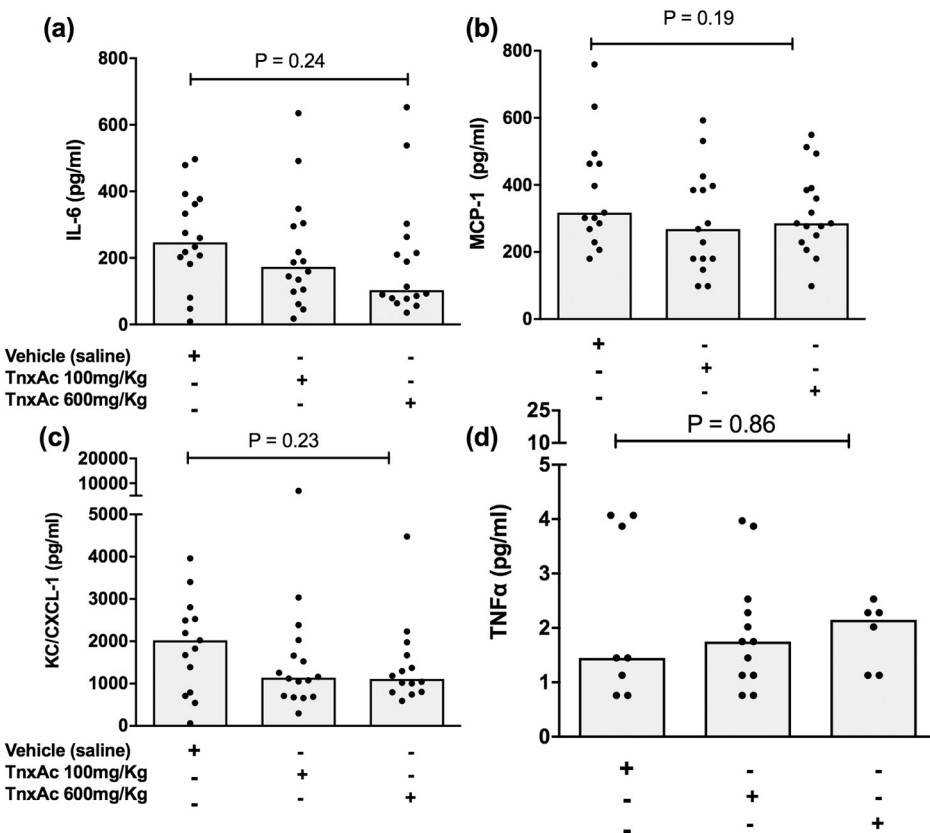

**Fig 5. Inflammatory cytokines and chemokines during polymicrobial sepsis.** Median serum levels of (a) IL-6, (b) MCP-1, (c) KC/CXCL-1 and (d) TNF-α of mice treated with TnxAc or vehicle measured 24 hours after sepsis induction; n = 6–16 per group; Kruskal-Wallis test.

studied. In a mouse model of *E. coli* peritonitis, t-PA deficiency has been associated with higher bacterial dissemination and decreased survival, while plasminogen deficiency behaved similar to wild type mice [29]. In contrast, in a *S. aureus* sepsis model, t-PA or plasminogen deficiencies were both associated with improved survival [30]. In a model of pulmonary sepsis caused by *Burkhoderia pseudomallei*, deficiency of fibrinolysis inhibitor PAI-1 was associated with increased bacterial spread and mortality [31], while deficiency of t-PA resulted in protection [32]. Similar results were obtained in mice with PAI-1 deficiency in *Haemophilus influenzae* and *Klebsiella pneumoniae* sepsis models [33,34]. Using the same polymicrobial sepsis used in our study, Shao et al demonstrated that mice with thrombin activatable fibrinolysis inhibitor (TAFI) deficiency presented decreased bacterial spread and improved survival [35]. Together, these results illustrate the heterogenous effect of fibrinolysis in sepsis, and support the concept that the fine balance between pro- and anti-fibrinolytic factors is an important element in the pathogenesis of this condition.

Based on these data we hypothesized that hypofibrinolysis induced by TnxAc could modulate the severity of sepsis, exerting a protective effect by limiting pathogen spread. We chose the polymicrobial sepsis model in order to evaluate pathogen spread from a discrete infectious site. Using two different doses of TnxAc, both capable to induce changes in hemostatic and fibrinolytic balance, we did not observe any protective effect of hypofibrinolysis during sepsis. Similarly, no difference could be observed in tissue damage and inflammation. While a mild non-significant trend towards lower bacterial burden in liver was observed, which reached

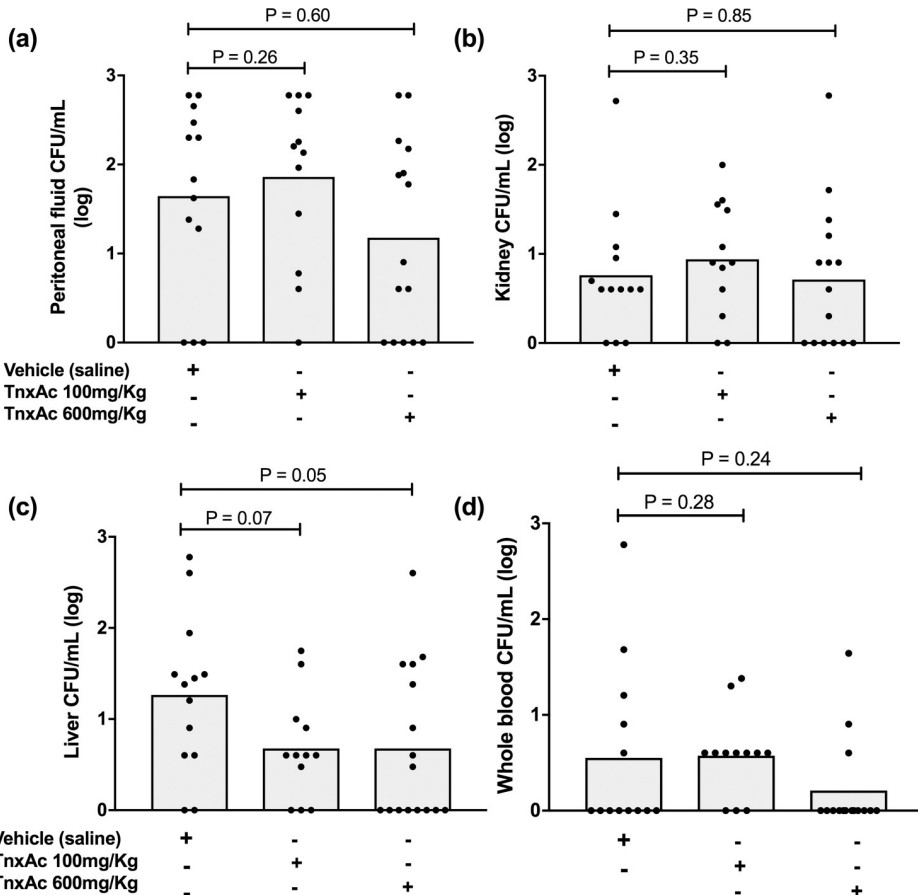

**Fig 6. Bacterial burden in TnxAc treated mice during polymicrobial sepsis.** Mean (log) counts of bacterial colonies per mL in (a) peritoneal fluid, (b) kidneys, (c) liver and (d) whole blood 24 hours after sepsis induction; n = 12–15 per group; Anova test with post-test correction.

statistical significance when all TnxAc-treated mice (100 mg/gk and 600 mg/kg) were compared with vehicle-treated mice, pathogen spread was not changed in whole blood, kidneys and peritoneal liquid. Considering the similar pathogen burden in these tissues, the marginal statistical significance in liver, and the inherent limitations of murine sepsis model, such as TnxAc doses and absence of antibiotic treatment, we believe that if at all present, any protective effect of TnxAc on bacterial dissemination should be of limited clinical relevance. Accordingly, since in our model hypofibrinolysis was present even before sepsis induction, as shown by our preliminary experiments with the TnxAc regimen that was used, our results demonstrate that hypofibrinolysis induced by this agent is compatible with a normal host response to infection and tissue damage.

Our study was initially designed to gain mechanistic insights into the role of fibrinolysis in the pathogenesis of sepsis, so that doses that are higher than the ones used in clinical practice were selected, based on previous studies with this agent in mice. However, given the recent increase in the use of TnxAc in clinical practice (mainly in conditions associated with higher thrombotic risk), the translation of our results into clinical practice could also be of interest. In this regard, one should note that therapeutic doses of TnxAc in humans (10 to 25 mg/kg/day) are lower than those used in our study, which represents a limitation to the translation of our results to humans. Yet, we believe that the 100mg/kg dose that we used in a group of mice is at

least closer to therapeutic doses used in humans (in fact, there are clinical reports of using doses of 100mg/kg [36]) than previous studies with TnxAc in mice models, which used doses ranging from 600mg/kg to 1,200mg/kg [18].

In humans, most situations in which TnxAc is used are associated with increased thrombotic risk such as elective orthopedic surgery [12], trauma [13] and puerperium [14]. Nonetheless, no evidence of increased thrombotic risk has been demonstrated in these trials. In our preliminary experiments usign a 48-hour TnxAc regimen in non-septic mice, thrombocytopenia was not observed, but TAT levels were mildly, but significantly increased, suggesting that the those dose regimens of TnxAc were sufficient to change the hemostatic balance towards a status of higher thrombin abundance. Of note, recent studies using TnxAc in therapeutic doses failed to show any procoagulant effect of this agent [37], although increased TAT levels have been described in TnxAc-treated patients in the past [38]. Since indirect markers of thrombin generation such as TAT and D-dimer are early and sensitive markers of DIC, and TnxAc has been shown to increase organ damage in murine endotoxemia [39], the observed increase in TAT levels in TnxAc-treated mice do not allow us to exclude that TnxAc could be deleterious during sepsis. However, the rate of microvascular thrombosis was not increased in TnxAc-treated mice 24 hours after polymicrobial sepsis, and that no clinically evident bleeding was observed in our 7-day survival experiments (data not shown). Together, while it is not possible to exclude that overt DIC could have been detected in later time-points, these clinical and histopathologic data suggest that overt DIC was not a major cause of death in our mice. In this regard, it should be noted that overt DIC was also not reported in the two previous studies that evaluated the effect of TnxAc in murine models of sepsis, even with higher doses of TnxAc. In the first of these two studies, which showed that a dose of 800mg/kg/dose every 8 hours increased mortality in a model of staphylococcal septic arthritis, no evidence of increased inflammation or coagulation activation could be demonstrated, so that authors speculated that TnxAc could be associated with non-hematological toxicity [17]. In a more recent study evaluating why TAFI deficiency was associated with improved survival in polymicrobial sepsis [35], mice were challenged with a dose of approximately 750mg/kg every 12 hours [16], which was associated with decreased survival in both knockout and wild type mice. Although authors from this latter study speculated that this effect could be due to thrombosis, no direct evidence of thrombi was shown, as this was not the objective of their study. Despite the lack of clinical and histopathological signs of overt DIC in our study, we would like to emphasize that our study was not designed to test the efficacy of TnxAc in sepsis, and do not support the use of this agent in these patients. As for the mechanisms responsible for increased TAT levels in TnxAc-treated mice, we speculate that this observation can be associated with the high/supratherapeutic doses of TnxAc used in our study, which could lead to alterations in the interplay between pro- and anti-thrombotic proteins, tipping the system towards a a mild procoagulant state. However, additional studies are necessary to elucidate this observation.

Our study has limitations that need to be acknowledged. First, we did not measure levels of specific mediators of fibrinolysis. However, we believe that the 12-hourly regimen of intraperitoneal TnxAc initiated 48h before sepsis and maintained up to 7 days (for survival curves), coupled with results from two laboratory assays capable to characterize changes in coagulation and fibrinolysis activation safely support that TnxAc-induced hypofibrinolysis was present, which was our goal (rather than demonstrate the kinetics of other mediators). A second limitation is the fact that markers of inflammation and coagulation activation were not measured at later time-points. The choice of a specific time-point for endpoint measurement in sepsis studies is always a difficult decision, and in our case it was based on our preliminary results indicating that mortality onset started around 48-hours after sepsis induction, as well as on CLP literature, which frequently assesses mice at earlier time-points. Accordingly, the 24-hour

time-point was considered a good compromise between capturing subclinical changes in inflammatory and biochemical biomarkers, and not excluding mice that evolved with a more severe sepsis phenotype, which could represent a bias to our study. Therefore, while we cannot rule out the possibility that different results would have been obtained at later time-points, we argue that if biologically relevant effects were present, at least part of their signals would be present at the 24-hour time-point. On the other hand, sampling animals at later time-points could have led to early mortality bias.

In conclusion, the crosstalk between hemostasis and inflammation during infections has been recently reconciled by the concept of immunothrombosis, which states that the host response to pathogens triggers a localized and transient thrombotic response that if regulated, facilitates pathogen clearance, but when exacerbated, could contribute to venous and arterial thrombosis [2]. Though negative and different that our initial hypothesis, our results add a new piece of data to this field, contributing to the understanding of the interplay between fibrinolysis activation and sepsis.

## Supporting information

**S1 Fig. Effect of TnxAc on coagulation and fibrinolysis parameters comparing mice treated with vehicle or with TnxAc (with both doses grouped together).**
(PDF)

**S2 Fig. Survival during polymicrobial sepsis.**
(PDF)

**S3 Fig. Biochemical markers of tissue damage comparing mice treated with vehicle or with TnxAc (with both doses grouped together).**
(PDF)

**S4 Fig. Inflammatory cytokines and chemokines during polymicrobial sepsis comparing mice treated with vehicle or with TnxAc (with both doses grouped together).**
(PDF)

**S5 Fig. Bacterial burden in TnxAc treated mice during polymicrobial sepsis comparing mice treated with vehicle or with TnxAc (with both doses grouped together).**
(PDF)

**S1 Table. Effect of TnxAc on hematological parameters (both doses grouped together).**
(PDF)

**S2 Table. Frequency of microvascular thrombosis (both doses grouped together).**
(PDF)

## Author Contributions

**Conceptualization:** Yzabella Alves Campos Nogueira, Erich Vinicius De Paula.

**Data curation:** Erich Vinicius De Paula.

**Formal analysis:** Yzabella Alves Campos Nogueira, Erich Vinicius De Paula.

**Funding acquisition:** Erich Vinicius De Paula.

**Investigation:** Yzabella Alves Campos Nogueira, Loredana Nilkenes Gomes da Costa.

**Methodology:** Yzabella Alves Campos Nogueira, Loredana Nilkenes Gomes da Costa, Carlos Emilio Levy, Fernanda Andrade Orsi.

**Project administration:** Yzabella Alves Campos Nogueira, Franciele de Lima, Erich Vinicius De Paula.

**Resources:** Yzabella Alves Campos Nogueira, Carlos Emilio Levy, Fernanda Andrade Orsi, Joyce M. Annichinno-Bizzacchi, Erich Vinicius De Paula.

**Supervision:** Erich Vinicius De Paula.

**Writing – original draft:** Yzabella Alves Campos Nogueira, Franciele de Lima, Erich Vinicius De Paula.

**Writing – review & editing:** Yzabella Alves Campos Nogueira, Loredana Nilkenes Gomes da Costa, Carlos Emilio Levy, Fernanda Andrade Orsi, Franciele de Lima, Joyce M. Annichinno-Bizzacchi, Erich Vinicius De Paula.

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
