## [Decision Letter · Decision Letter 0]

11 Oct 2019

PONE-D-19-19607

Hypofibrinolysis induced by tranexamic acid does not influence inflammation and mortality in a polymicrobial sepsis model

PLOS ONE

Dear Dr. De Paula,

Thank you for submitting your manuscript to PLOS ONE. After careful consideration, we feel that it has merit but does not fully meet PLOS ONE’s publication criteria as it currently stands. Therefore, we invite you to submit a revised version of the manuscript that addresses the points raised during the review process.

The article has been reviewed by two experts. They consider that the study is sound and conclusions are supported by the experimental data. However, some of the statements in the text do not seem to be in accordance with the data. In particular, in relation to the role of coagulation and fibrinolysis with infectivity. The increase in TAT is significant and has to be discussed in the context of the results and those in the literature.

We would appreciate receiving your revised manuscript by Nov 25 2019 11:59PM. To enhance the reproducibility of your results, we recommend that if applicable you deposit your laboratory protocols in protocols.io, where a protocol can be assigned its own identifier (DOI) such that it can be cited independently in the future. For instructions see: http://journals.plos.org/plosone/s/submission-guidelines#loc-laboratory-protocols

We look forward to receiving your revised manuscript.

Kind regards,

Pablo Garcia de Frutos

Academic Editor

PLOS ONE

Journal Requirements:

Reviewers' comments:

Reviewer's Responses to Questions

**Comments to the Author**

1. Is the manuscript technically sound, and do the data support the conclusions?

Reviewer #1: Yes

Reviewer #2: Yes

2. Has the statistical analysis been performed appropriately and rigorously? 

Reviewer #1: Yes

Reviewer #2: Yes

3. Have the authors made all data underlying the findings in their manuscript fully available?

Reviewer #1: Yes

Reviewer #2: Yes

4. Is the manuscript presented in an intelligible fashion and written in standard English?

Reviewer #1: Yes

Reviewer #2: Yes

5. Review Comments to the Author

Reviewer #1: The authors describe findings on an interesting and clinically relevant topic. The following remarks should be adressed in a revised MS:

- The TXA-dosage is considerably higher than that in clinical practice (100/600 mg per kg vs. 15-25 mg per kg). It should be explicitely discussed that this difference could negatively influence translation of theses findings on the clinical situation.

- The numbers of mice in all groups should be mentioned in the methods section.

- When comparing the clinical score (M-CASS), which should be described im more detail in the methods section, it seems to be considerably higher in the TXA-Groups than in vehicle. The results of statistical comparisons should be presented.

- Figure 5a indicates lower IL6 in TXA-treated animals compared to vehicle. I suggest to add a cut-off based chi2-testing. If you categorize be < vs. >/= 100 pg/ml, TXA-treatment clearly seems to be associated with lower IL6-levels.

- I interprete figure 6c that there is a trend for lower liver CFU. If this is the case, the heading (lines 241-2) and the corresponding section (243-5) doesn't agree with each other. This should be rephrased.

Finally, for all testings comparisons between vehicle vs. (TXA low AND TXA high) are necessary. As the single TXA-groups are smaller than vehicle this would improve the sample number.

Reviewer #2: This is a limited-scope study testing the effect of pharmacologic suppression of fibrinolysis on the outcome of murine polymicrobial, focal sepsis in the CLP model. The key parameters measured (fibrinolysis via euglobulin time, survival, bacterial dissemination, 7-day survival) support the overall conclusion that there is no significant effect of Tx.

On the other hand, there are recognizable trends for reduced dissemination in liver and whole blood, as well as statistically significant TAT level increases. The latter observation somewhat runs counter to the stated conclusion that there is no increased thrombosis / DIC risk.

The authors should also attempt to provide a mechanistic explanation for the increased TAT (which reflects thrombin abundance)

Neither body text data shown in figure 6 do not support the header of the section "Hypofibrinolysis is associated with lower bacterial dissemination. The header for this paragraph should be: Hypofibrinolysis is NOT associated with bacterial dissemination.

Aside from these minor comments, the study makes a minor contribution to an already substantial body on the role of fibrinolyis in various models of murine sepsis/infection.

6. PLOS authors have the option to publish the peer review history of their article (what does this mean?). If published, this will include your full peer review and any attached files.

Reviewer #1: No

Reviewer #2: No

---

## [Author Response · Author response to Decision Letter 0]

5 Dec 2019

Response to Reviewers 

Manuscript title: “Hypofibrinolysis induced by tranexamic acid does not influence inflammation and mortality in a polymicrobial sepsis model”

Reviewer #1:

We are grateful for the reviewer’s comments and inquiries, and we sincerely appreciate your effort to improve our manuscript. 

Obs: line numbers might change in the version in which changes are tracked (for which lines are referred to) or accepted. We apologize for that. 

Reviewer comments: 

1. The TXA-dosage is considerably higher than that in clinical practice (100/600 mg per kg vs. 15-25 mg per kg). It should be explicitely discussed that this difference could negatively influence translation of these findings on the clinical situation.

We agree with the need to emphasize the effects of this dose discrepancy in the translation of our results to human biology. When we first looked in the literature to decide which dose to use we were surprised that in mice studies, tranexamic acid was used in much higher doses than in humans (as high as 1,200mg/kg), and we did not find any study that used the 15-25mg/kg. Therefore, we chose the 600mg/kg dose which is the same order of magnitude of previous mice studies, so that our results could be comparable with mice studies. However, we also wanted to use a dose that was closer to clinically used doses, and this is why we chose the 100mg/kg (there are some reports of TnxAc used in this dose). Still, the difference is high and we agree that it could negatively influence translation to clinical situations. So, in the revised version we explicitly mention this limitation by rewriting one of the paragraphs of the discussion. 

Discussion lines 326-337.

“Our study was initially designed to gain mechanistic insights into the role of fibrinolysis in the pathogenesis of sepsis, so that doses that are higher than the ones used in clinical practice were selected, based on previous studies with this agent in mice. However, given the recent increase in the use of TnxAc in clinical practice (mainly in conditions associated with higher thrombotic risk), the translation of our results into clinical practice could also be of interest. In this regard, one should note that therapeutic doses of TnxAc in humans (10 to 25 mg/kg/day) are lower than those used in our study, which represents a limitation to the translation of our results to humans. Yet, we believe that the 100mg/kg dose that we used in a group of mice is at least closer to therapeutic doses used in humans (in fact, there are clinical reports of using doses of 100mg/kg [36]) than previous studies with TnxAc in mice models, which used doses ranging from 600mg/kg to 1,200mg/kg [18].” 

2. - The numbers of mice in all groups should be mentioned in the methods section.

Thank you for the suggestion. We included this information in the methods section, along with an explanation of how these numbers were calculated. We also updated figure 1 legend (line 192) in which this information was missing, since this information is available in all other figure legends. 

Methods, line 172-176

“The number of mice used in each experiment was calculated to obtain a power of 80% with a type I error of 0.05 to detect differences of at least two standard deviations, and ranged from 13 to 29 mice per group in survival curves, 6 to 11 in hemostasis assays, 5 to 12 in biochemical assays, 6 to 16 for inflammatory markers, and 12 to 15 in bacterial burden experiments.”

3. When comparing the clinical score (M-CASS), which should be described im more detail in the methods section, it seems to be considerably higher in the TXA-Groups than in vehicle. The results of statistical comparisons should be presented.

The scoring system was detailed in the methods section: 

Methods, line 100-103: 

“… murine sepsis (M-CASS), which considers the following markers: fur aspect, activity, posture, behavior, chest movements, chest sounds, and eye lid opening, ranked form 1 to 4 [19].” 

The statistical results were checked and we confirmed that there are no significant differences between groups. The test is the Kruskall-Wallis and the post test reveals no difference between time-points comparing the three groups, with the higher P value of 0.6). However, while analyzing why in the previous figure this lack of difference was not clear, we noticed that mice that had reached the highest score were left in the dataset with highest possible score until day 7, after they died. In the revised version, we only plotted the score from mice that were alive at each time point. The statistical analysis did not change, but we believe that now the figure conveys a fairer view of the results. The legend was changed to explain this. 

Revised figure 3 and legend (lines 212, 213, 215, 216):

In the attached document, we embedded revised figure 3

Figure 3. Clinical score of the murine experimental sepsis. Bars indicate daily mean clinical scores (and SEM) of mice treated with TnxAc at 100 mg/Kg, 600 mg/kg or vehicle (each bar represents one day of the follow-up, with data from mice that were remained alive at each time-point). No statistically significant difference could be observed between groups for each day separately (Kruskall-Wallis test with Dunn’s multiple comparison test, comparing each time point between the three experimental groups). 

4. Figure 5a indicates lower IL6 in TXA-treated animals compared to vehicle. I suggest to add a cut-off based chi2-testing. If you categorize be < vs. >/= 100 pg/ml, TXA-treatment clearly seems to be associated with lower IL6-levels.

We thank you for the suggestion. We performed this analysis but still differences did not reach statistical significance, with the comparison of vehicle x 100mg/kg yielding a p value of 0.6689, and vehicle versus 600mg/kg a p value of 0.06. In addition, when all mice treated with TnxAc (both groups) were compared with vehicle, as suggested below, the P value actually increased to 0.11. Therefore, we chose not to emphasize this specific trend in the results section. Yet, we specify now that the section heading refers to “statistically significant differences”, and the overall issue of trends was discussed in the Discussion (see item 5). The fact that other markers were not altered influenced this decision. 

The following terms were included in data from figure 5 description: 

Results, line 239:

“No statistically significant difference could be observed in levels of inflammatory mediators in plasma of mice treated with TnxAc at either dose, when compared to vehicle-treated mice (Fig 5).” 

5. I interprete figure 6c that there is a trend for lower liver CFU. If this is the case, the heading (lines 241-2) and the corresponding section (243-5) doesn't agree with each other. This should be rephrased.

The issue of how we should manage trends here comes up again, but this time we agree with the reviewer that there is enough data to support a less conservative interpretation. First, the P value is really nearly significant (P=0.05). Second, when we grouped TnxAc treated mice together (suggestion in item 6 from this reviewer), the P reached statistical significance (<0.04). Yet, the pre-determined threshold for significance was not reached. In order to accommodate these data and aspects, the following changes were included:

Results, lines 248-255

“Hypofibrinolysis induced by TnxAc is not associated with significant decreases of bacterial dissemination

Bacterial clearance was studied in whole blood, peritoneal fluid, liver and kidneys. No statistically significant differences were observed in bacterial load in any of these tissues, at either tested doses (Fig 6). Yet, we could observe a trend towards lower bacterial burden in liver of mice treated with the highest dose of TnxAc (P=0.05) (Fig 6c), which reached statistical significance when all TnxAc-treated mice were compared with vehicle-treated mice (supplementary figure 5c).”

6. Finally, for all testings comparisons between vehicle vs. (TXA low AND TXA high) are necessary. As the single TXA-groups are smaller than vehicle this would improve the sample number.

We performed all analyses grouping both TnxAc doses. Results did not change in survival studies, nor in hematology, microvascular thrombosis, biochemistry or cytokine. The marginal p value observed in liver CFU counts decreased a little and reached statistical significance, suggesting that TnxAc could be associated with a lower bacterial burden in liver. All results are cited in the results section, and presented as a supplementary data file that was added to the manuscript. We were cautious in interpreting trends, and since the results did not reach our predetermined p threshold for significance, we believe we should not change the main conclusion based on them. Yet, we agree that this trend could indicate a mild protective effect of TnxAc in bacterial spread, and this was discussed in the revised version. 

Discussion, lines 269-272

“Our main results were that hypofibrinolysis induced by TnxAc does not affect the clinical course of sepsis nor its subclinical markers of tissue damage and inflammation, and is not associated with statistically significant changes in pathogen spread.”

Discussion, lines 313-322

“Similarly, no difference could be observed in tissue damage and inflammation. While a mild non-significant trend towards lower bacterial burden in liver was observed, which reached statistical significance when all TnxAc-treated mice (100 mg/gk and 600 mg/kg) were compared with vehicle-treated mice, pathogen spread was not changed in whole blood, kidneys and peritoneal liquid. Considering the similar pathogen burden in these tissues, the marginal statistical significance in liver, and the inherent limitations of murine sepsis model, such as TnxAc doses and absence of antibiotic treatment, we believe that if at all present, any protective effect of TnxAc on bacterial dissemination should be of limited clinical relevance. Accordingly, since in our model hypofibrinolysis was present even before sepsis induction, as shown by our preliminary experiments with the TnxAc regimen that was used, our results demonstrate that hypofibrinolysis induced by this agent is compatible with a normal host response to infection and tissue damage.”

Reviewer #2:

We thank you for the all the suggestions and comments to enhance our manuscript.

Obs: line numbers might change in the version in which changes are tracked (for which lines are referred to) or accepted. We apologize for that. 

Reviewer comments: 

1. This is a limited-scope study testing the effect of pharmacologic suppression of fibrinolysis on the outcome of murine polymicrobial, focal sepsis in the CLP model. The key parameters measured (fibrinolysis via euglobulin time, survival, bacterial dissemination, 7-day survival) support the overall conclusion that there is no significant effect of Tx.

Thank you for your comments. We agree that our results do not allow us to claim that TnxAc has a significant effect on sepsis parameters that were analyzed in this model, and we tried to convey this message, avoiding to overinterpret statistical trends.

2. On the other hand, there are recognizable trends for reduced dissemination in liver and whole blood, as well as statistically significant TAT level increases. The latter observation somewhat runs counter to the stated conclusion that there is no increased thrombosis / DIC risk.

Thank you for the comments. We agree that the results of liver CFU count represent a trend, which should be discussed. Reviewer 1 also suggested that we included a new set of analyses, comparing all TnxAc-treated mice (both doses) with vehicle treated mice. These data were included as supplementary data and the only result that changed was precisely the bacterial burden in liver, which reached statistical significance (for whole blood, the trend, which was already very mild, disappeared, as shown in supplementary figure 5). So, we included a discussion about the liver results in the revised version. However, since our predetermined significance level was not reached, we refrained from changing the overall conclusion that TnxAc protects from bacterial dissemination. We tried to use conservative language presenting all the data. 

In regard to the second part of this query (the TAT issue) in our response is in query #3 (below).

Discussion, lines 269-272

Our main results were that hypofibrinolysis induced by TnxAc does not affect the clinical course of sepsis nor its subclinical markers of tissue damage and inflammation, and is not associated with statistically significant changes in pathogen spread. 

Discussion, lines 313-322

“Similarly, no difference could be observed in tissue damage and inflammation. While a mild non-significant trend towards lower bacterial burden in liver was observed, which reached statistical significance when all TnxAc-treated mice (100 mg/gk and 600 mg/kg) were compared with vehicle-treated mice, pathogen spread was not changed in whole blood, kidneys and peritoneal liquid. Considering the similar pathogen burden in these tissues, the marginal statistical significance in liver, and the inherent limitations of murine sepsis model, such as TnxAc doses and absence of antibiotic treatment, we believe that if at all present, any protective effect of TnxAc on bacterial dissemination should be of limited clinical relevance. Accordingly, since in our model hypofibrinolysis was present even before sepsis induction, as shown by our preliminary experiments with the TnxAc regimen that was used, our results demonstrate that hypofibrinolysis induced by this agent is compatible with a normal host response to infection and tissue damage.”

3. The authors should also attempt to provide a mechanistic explanation for the increased TAT (which reflects thrombin abundance)

We thank the reviewer for pointing out both the fact that higher TAT levels are counter the argument that DIC was not elicited in our model, as well as for the need for a mechanistic explanation. We did not find any data on the effect of TnxAc in thrombin generation in the dose-range (supratherapeutic doses) that were used in our study. Recent studies in humans failed to show increases in thrombin generation in TnxAc-treated patients (cited in the revised version), although we were able to find a report of higher TAT levels associated with TnxAc in humans (also cited). So, in regard to whether this increase should be viewed as a sign of increased risk of DIC, we agree that a more conservative language should be used, particularly because assessing the safety of TnxAc was not the objective of our study. So, in addition to introducing a paragraph discussing these issues, we also excluded all statements about safety of TnxAc from the revised version. We believe that this reviewer comment allowed a major contribution to improve our manuscript, since we agree that we were maybe too liberal in addressing TnxAc safety from our data. In regard to the mechanistic explanation, we can only speculate that higher TAT levels are more likely specific of the supratherapeutic doses of TnxAc (for which there is little to no clinical experience), and that prolonged and intense fibrinolysis inhibition could lead to alterations in the interplay between pro and anti-thrombotic proteins, tipping the system towards increased thrombin generation. However, as acknowledged in the revised version, additional studies are needed to explore this finding. 

Discussion, lines 345-360

“In our preliminary experiments usign a 48-hour TnxAc regimen in non-septic mice, thrombocytopenia was not observed, but TAT levels were mildly, but significantly increased, suggesting that the those dose regimens of TnxAc were sufficient to change the hemostatic balance towards a status of higher thrombin abundance. Of note, recent studies using TnxAc in therapeutic doses failed to show any procoagulant effect of this agent [37], although increased TAT levels have been described in TnxAc-treated patients in the past [38]. Since indirect markers of thrombin generation such as TAT and D-dimer are early and sensitive markers of DIC, and TnxAc has been shown to increase organ damage in murine endotoxemia [39], the observed increase in TAT levels in TnxAc-treated mice do not allow us to exclude that TnxAc could be deleterious during sepsis. However, the rate of microvascular thrombosis was not increased in TnxAc-treated mice 24 hours after polymicrobial sepsis, and that no clinically evident bleeding was observed in our 7-day survival experiments (data not shown). Together, while it is not possible to exclude that overt DIC could have been detected in later time-points, these clinical and histopathologic data suggest that overt DIC was not a major cause of death in our mice. In this regard, it should be noted that overt DIC was also not reported in the two previous studies that…”

Discussion, lines 377-385

“Despite the lack of clinical and histopathological signs of overt DIC in our study, we would like to emphasize that our study was not designed to test the efficacy of TnxAc in sepsis, and do not support the use of this agent in these patients. As for the mechanisms responsible for increased TAT levels in TnxAc-treated mice, we speculate that this observation can be associated with the high/supratherapeutic doses of TnxAc used in our study, which could lead to alterations in the interplay between pro- and anti-thrombotic proteins, tipping the system towards a a mild procoagulant state. However, additional studies are necessary to elucidate this observation.”

EXCLUDED

Discussion, line 422

 “In addition, we provide further support to the thrombotic safety of standard doses of TnxAc, a low cost and widely used agent to prevent bleeding.”

Abstract, line 40

“While our results do not support the use of TnxAc in sepsis, they also address the thrombotic safety of TnxAc, a low cost and widely used agent to prevent bleeding.“

* “also address” was previously, “corroborated”.

4. Neither body text data shown in figure 6 do not support the header of the section "Hypofibrinolysis is associated with lower bacterial dissemination. The header for this paragraph should be: Hypofibrinolysis is NOT associated with bacterial dissemination.

We thank you for noticing this error, which was corrected in the revised version, and the heading now reads “Hypofibrinolysis induced by TnxAc is not associated with significant decreases of bacterial dissemination”. This was actually our conclusion, as evidence by several statements throughout the manuscript. So, we excluded the “not” and changed the sentence ending so as to accommodate the acknowledgement that while there were no statistically significant differences, there was a trend worth discussing. 

5. Aside from these minor comments, the study makes a minor contribution to an already substantial body on the role of fibrinolyis in various models of murine sepsis/infection.

We thank you for the comments which certainly improved the manuscript. We agree that the study represents a minor contribution, yet an original piece of information within this important area of research. 

Sincerely Yours, 

Erich V De Paula, MD, PhD

University of Campinas

Corresponding author

Email: erich@unicamp.br

Phone: +55 19 996050301

---

## [Editor Report · Decision Letter 1]

10 Dec 2019

Hypofibrinolysis induced by tranexamic acid does not influence inflammation and mortality in a polymicrobial sepsis model

PONE-D-19-19607R1

Dear Dr. De Paula,

We are pleased to inform you that your manuscript has been judged scientifically suitable for publication and will be formally accepted for publication once it complies with all outstanding technical requirements.

With kind regards,

Pablo Garcia de Frutos

Academic Editor

PLOS ONE
---

## [Editor Report · Acceptance letter]

20 Dec 2019

PONE-D-19-19607R1 

Hypofibrinolysis induced by tranexamic acid does not influence inflammation and mortality in a polymicrobial sepsis model 

Dear Dr. De Paula:

I am pleased to inform you that your manuscript has been deemed suitable for publication in PLOS ONE. Congratulations! Your manuscript is now with our production department. 

With kind regards,

on behalf of

Dr. Pablo Garcia de Frutos 

Academic Editor

PLOS ONE